# Identification of potential target genes and crucial pathways in small cell lung cancer based on bioinformatic strategy and human samples

Xiuwen Chen[1], Li Wang[1], Xiaomin Su[2], Sen-yuan Luo[1], Xianbin Tang[1], Yugang Huang◉[1]*

**1** Department of Pathology, Taihe Hospital, Hubei University of Medicine, Hubei, China, **2** Department of Immunology, Nankai University School of Medicine, Tianjin, China

* huangyg2018@outlook.com

**Data Availability Statement:** All publicly available datasets analyzed in this study can be found here: https://www.ncbi.nlm.nih.gov/geo/. GEO accession

## Abstract

Small cell lung cancer (SCLC) is a carcinoma of the lungs with strong invasion, poor prognosis and resistant to multiple chemotherapeutic drugs. It has posed severe challenges for the effective treatment of lung cancer. Therefore, searching for genes related to the development and prognosis of SCLC and uncovering their underlying molecular mechanisms are urgent problems to be resolved. This study is aimed at exploring the potential pathogenic and prognostic crucial genes and key pathways of SCLC via bioinformatic analysis of public datasets. Firstly, 117 SCLC samples and 51 normal lung samples were collected and analyzed from three gene expression datasets. Then, 102 up-regulated and 106 down-regulated differentially expressed genes (DEGs) were observed. And then, functional annotation and pathway enrichment analyzes of DEGs was performed utilizing the FunRich. The protein-protein interaction (PPI) network of the DEGs was constructed through the STRING website, visualized by Cytoscape. Finally, the expression levels of eight hub genes were confirmed in Oncomine database and human samples from SCLC patients. It showed that *CDC20*, *BUB1*, *TOP2A*, *RRM2*, *CCNA2*, *UBE2C*, *MAD2L1*, and *BUB1B* were upregulated in SCLC tissues compared to paired adjacent non-cancerous tissues. These suggested that eight hub genes might be viewed as new biomarkers for prognosis of SCLC or to guide individualized medication for the therapy of SCLC.

## Introduction

Lung cancer is a subtype of malignant tumors with a peak risk of morbidity and mortality, which makes it a notable healthcare issue for human beings. In 2018, the total number of cases newly diagnosed as lung cancer was about 2.09 million (11.6% of all newly diagnosed cancers), and the number of new deaths was about 1.76 million (18.4% of all sites) worldwide [1]. Small cell lung cancer (SCLC) accounts for approximately 20% of lung cancer patients and belongs to neuroendocrine tumor [2]. Different from other histopathological subtypes of lung cancer,

numbers for the data sets used in our study included GSE40275, GSE99316, and GSE99316.

**Funding:** This work was supported by National Natural Science Foundation of China (grant number 81600436 and 81974088). The funders had no role in study design, data collection and analysis, decision to publish, or preparation of the manuscript.

**Competing interests:** The authors have declared that no competing interests exist.

SCLC is accompanied by rapid clinical progression. Almost all patients with SCLC have extensive metastasis when diagnosed, and it has a poor prognosis. The 5-year relative overall survival (OS) rate is not more than 6%. In clinical practice, conventional treatments of SCLC include chemotherapy, radiotherapy, surgery and immunotherapy. Chemotherapy for SCLC is the main treatment currently, but there are still many problems such as drug resistance and easy relapse [3, 4]. In the past few decades, the survival rate of SCLC patients has not been evidently improved, and no molecular-targeted drugs have been shown to significantly prolong the survival time of patients [5]. In recent years, studies have been reported that many molecular mechanisms altered in SCLC including induced expression of oncogene, such as *MYC* [6] and *FGFR1* [7], and deletion of tumor-suppressor genes, such as *TP53*, *PTEN*, *RB*, and *FHIT* [8]. Changes in these related genes and signaling pathways promote cell proliferation and inhibition of apoptosis, leading to early-stage metastasis of tumor cells, such as mutation, methylation or expression of *PIK3CA*, *PTEN*, *AKT* and other genes in the *PI3K/AKT/mTOR* pathway [9].

Because of the complexity of biological characteristics and poor prognosis of SCLC, the key biomarkers and specific targets for occurrence and development of SCLC are not well known. Therefore, it is necessary to explore more genetic information to screen out potential or promising biomarkers for early-stage diagnosis and precision medical treatment of SCLC. In recent years, gene chip technology and bioinformatics analysis has been widely used to identify molecular changes in tumorigenesis and development and has been proved to be an efficient method for identifying key genes in the research of genomics [10–12]. However, due to the strong invasion and short life span of SCLC, the data of related gene chips are infrequent.

In this study, gene expression data obtained from GEO databases were integrated to conduct data mining and analysis of SCLC. Then, a series of co-differentially expressed genes have been screened in SCLC. A series of analysis were carried out based on these genes, including analysis of functional enrichment, protein-protein interaction network and human samples validation. We identified numbers of hub genes analyzed the interaction between genes and drugs. Our research may offer more insight into the molecular mechanisms or study of available drugs for this epidemic and destructive disease. The workflow for bioinformatics strategy of SCLC was illustrated in Fig 1.

## Materials and methods

### SCLC gene expression data from GEO data repository

As a publicly genomics database, Gene Expression Omnibus (GEO) of NCBI (https://www.ncbi.nlm.nih.gov/) collects submitted high-throughput gene expression data and can be used for retrieving all datasets involving studies of SCLC. For our following analysis, it was considered reasonable for studies that met the following criteria: (1) Studies of human SCLC and corresponding adjacent or normal lung tissues. (2) There are detailed information of research technology and platform. (3) All studies have been published in English. Based on these criteria, three gene expression microarray datasets for SCLC, including GSE40275, GSE99316, and GSE60052, were taken from GEO. Details of these microarray studies were shown in Table 1.

### Data preprocessing and DEGs screening

The DEGs between SCLC and normal lung samples from GEO datasets, GSE40275 and GSE99316, were screened by using GEO2R (http://www.ncbi.nlm.nih.gov/geo/geo2r). GEO2R is an online analysis tool for comparing two or more groups of data in the GEO series to identify DEGs under the same detection method. The matrix data (.TSV, from supplementary files) of GSE60052 were normalized and $log_2$ transformed by limma package of R [13]. The

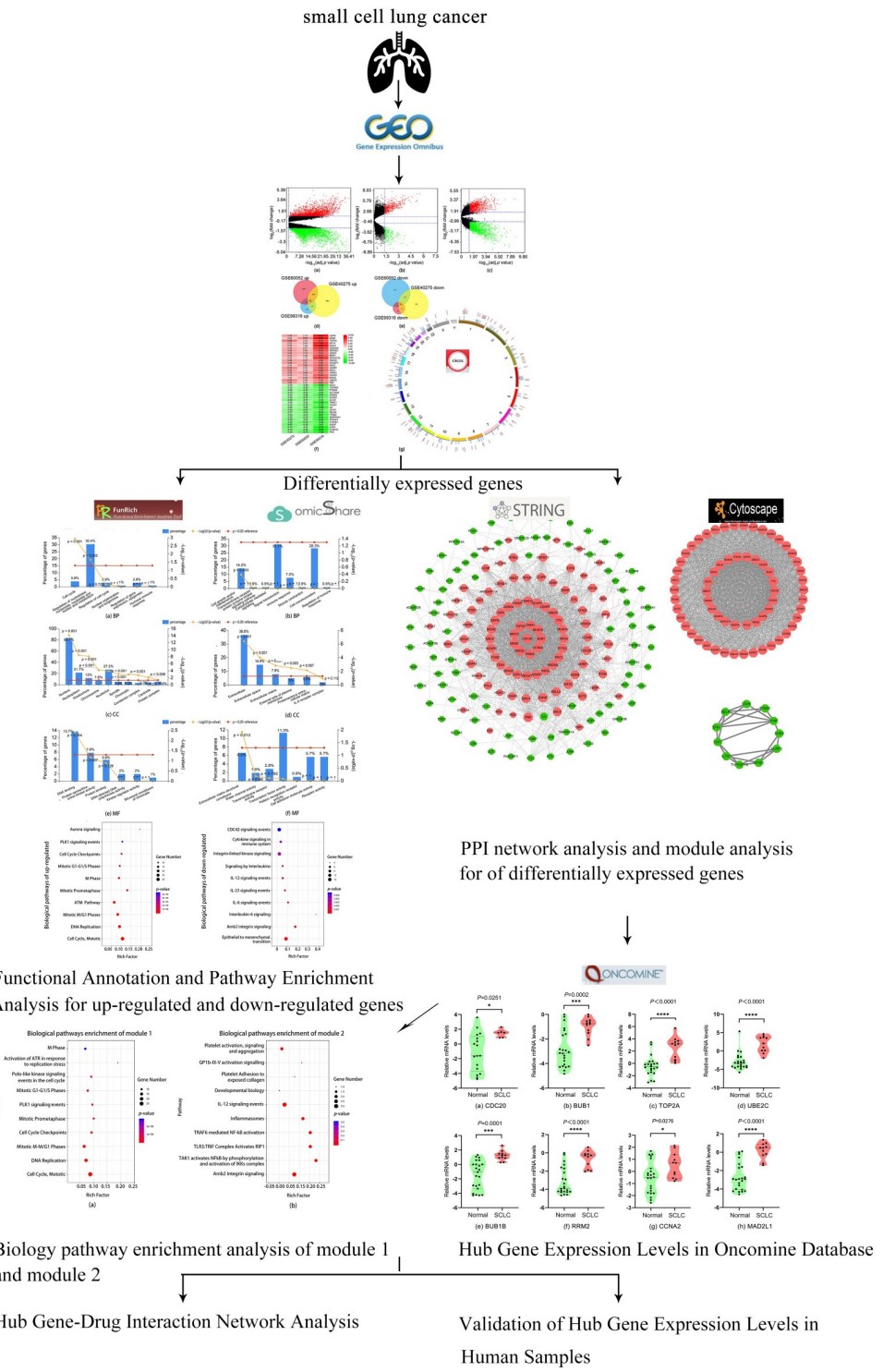

**Fig 1. Workflow for bioinformatics strategy of key genes and biological pathways in SCLC.**

**Table 1. Details of associated microarray datasets from GEO database in SCLC studies.**

| GSE | Reference PMID | Platform | Sample | | Tumor type | Country | Lastest update |
|---|---|---|---|---|---|---|---|
| | | | Normal lung | Tumor | | | |
| GSE40275 | None | GPL15974 Human Exon 1.0 ST Array [CDF: Brainarray Version 9.0.1, HsEx10stv2_Hs_REFSEQ] | 43 | 15 | SCLC | Austria | 2012 |
| GSE99316 | 23714854 | GPL570 [HG-U133_Plus_2] Affymetrix Human Genome U133 Plus 2.0 Array | 1 | 23 | SCLC | Japan | 2019 |
| GSE60052 | 27093186 | GPL11154 Illumina HiSeq 2000 (Homo sapiens) | 7 | 79 | SCLC | USA | 2019 |

SCLC, small cell lung cancer.

application of the adjusted *P*-value (adj. *P*) can balance the detection of statistically significant genes and the restriction of false positive. Probe sets without gene symbols were removed, and genes with multiple probe sets were averaged. Next, |log2FC (fold change) |>1 and adj. *P*-value<0.05 was considered to be statistically significant. To visualize the identified DEGs, SangerBox (soft.sangerbox.com) and FunRich [14] were used to make volcano plots and Venn diagrams, respectively.

## Circular visualization of the DEGs

Circos (http://circos.ca/) was applied to display our data for a better understanding of DEGs, including their gene symbols and locations on chromosomes [15].

## Functional enrichment analysis of DEGs

Funrich is a publicly available software for Functional annotation and pathway enrichment analysis of genes or proteins. In this study, Funrich was used to analyze the functional enrichment of up-regulated and down-regulated DEGs, including molecular function, biological process, cell composition and biological pathway. The results of the biological pathway analysis were visualized by OmicShare tools (http://www.omicshare.com/tools), a free online platform for data analysis.

## Construction and analysis of protein-protein interaction (PPI) network

For the better view of the relationship among these DEGs, the PPI network was constructed via using The Search Tool for the Retrieval of Interacting Genes (STRING) database (version 11.0; http://string-db.org/) [16]. In the present study, Cytoscape software (version 3.6.1) was used to establish and visualize PPI networks [17]. As one of the characteristics of PPI network, the network module contained specific biological significance. The salient modules in this PPI network were explored by The Cytoscape plug-in Molecular Complex Detection (MCODE) [18]. The thresholds were set as follows: Degree cutoff = 2, Node Score Cutoff = 0.2, and K-Core = 2. Next, the pathway enrichment analysis of the DEGs in different modules was performed by FunRich and visualized by the OmicShare tools.

## Screening of hub genes

The hub genes in the PPI network were screened by cytohubba, a plug-in of Cytoscape software (version 3.6.1). The genes with degree score ≥75 were considered as hub genes.

## Verification of hub genes expression levels

At first, the expression of these hub genes was validated in Oncomine database. As one of the world's largest tumor gene microarray database and integrated data analysis platform, Oncoline (oncomine.org) is designed to excavate cancer genetic information. The database hitherto has collected data from 729 gene expression data sets and over 90,000 cancer and normal tissue samples. It can be used to uncover the differential expression of a single gene in SCLC tissue and its related normal tissues [19]. To figure out the expression levels of hub genes in SCLC, SCLC gene expression data from the study of Garber et al [20] and Bhattacharjee et al [21] in the Oncomine database were investigated and visualized by GraphPad Prism. Thresholds for the data type was restricted to mRNA. Then, the expression levels of hub genes were further verified by quantitative real-time PCR (RT-qPCR) through tissue samples of SCLC patients and paired adjacent non-cancerous ones.

## Construction and analysis of hub gene-drug interaction network

The hub gene-drug interaction networks were constructed by Comparative Toxicogenomics Database (CTD) [22], a platform for analysis chemotherapeutic drugs which could inhibit or induce the mRNA or protein expression of hub genes. The hub gene-drug interaction was investigated in CTD database and visualized by the OmicShare tools.

## Human SCLC samples

All 7 SCLC tissues and 7 paired adjacent non-cancerous tissue samples of PPFE (Formalin-fixed and paraffin embedded) were collected from patients who had been diagnosed as SCLC from May 2019 to May 2020 at Taihe Hospital of Hubei University of Medicine, China. The PPFE samples were stored at room temperature until total RNA was extracted. All SCLC patients were diagnosed and graded according to the pathological characteristics in the Department of Pathology, Taihe Hospital. All human samples were obtained by informed consent (IFC) from patients, and this study was supported and approved by by the Ethics Committee of Taihe Hospital.

## RNA isolation and reverse transcription PCR

Quantitative real-time PCR (RT-qPCR) were performed according to methods published previously [23, 24]. Total RNA was extracted from the PPFE samples by using RNeasy FFPE Kit (cat. no. 73504, QIAGEN, Germany). Total RNA of 1 μg was reversely transcribed in a 20 μl reaction using RevertAid First Strand cDNA Synthesis Kit (cat. no. #K1622, Thermo Scientific, USA) according to the manufacturer's protocol. The reaction products were then diluted with 40 μl RNase-free water. The real-time PCR reaction was composed of 2 μl cDNA, 10 μl of PowerUp™ SYBR™ Green Master Mix (cat. no. A25741, Thermo Scientific, USA) and 0.5 μl of forward and reverse primers (0.5 μM). RT-qPCR was conducted in an ABI Prism 7500 analyzer (Applied Biosystems, USA) for 40 cycles (95˚C for 15 sec, 58˚C for 15 sec, 72˚C for 30 sec) after an initial 120s denaturation at 95˚C. HPRT1 was endogenous reference gene. All reactions were run in triplicate. The relative RNA levels of SCLC samples were calculated by using the $2^{-\Delta\Delta Ct}$ method. All primers of the hub genes and HPRT1 were synthesized by Sangon Biotech (Shanghai, China), and the information of their sequences were listed in Table 2.

## Statistical analysis

Statistical analysis was performed through GraphPad Prism (version 8.2.1, San Diego, CA) software. Student's t-tests were used for the comparison of two sample groups. Differences

**Table 2. RT-PCR primers of 8 most significant hub genes and reference gene, HPRT1.**

| Gene name | Full name | Primer sequence(5'→3') | Tm(˚C) | Product(bp) |
|---|---|---|---|---|
| CDC20 | cell division cycle 20, aliases: CDC20A | Forward: AGTTCGCGTTCGAGAGT | 55.7 | 195 |
| | | Reverse: GAACCTTGGAACTGGAT | 50 | |
| BUB1 | budding uninhibited by benzimidazoles-1, aliases: BUB1A | Forward: TATAGCAGGCTGATTGGGCT | 58.5 | 107 |
| | | Reverse: TGGCTTAAACAGGTCAGTGT | 57 | |
| TOP2A | Topoisomerase II alpha | Forward: GTGTCACCATTGCAGCCTGT | 61 | 152 |
| | | Reverse: GAACCAATGTAGGTGTCTGG | 56 | |
| UBE2C | ubiquitin conjugating enzyme E2 C | Forward: CTCATGGTATATGAAGACCT | 51 | 132 |
| | | Reverse: GCATATGTTACCCTGGGTGT | 57 | |
| BUB1B | BUB1 mitotic checkpoint serine/threonine kinase B | Forward: TGCTCTGAGTGAAGCCATGT | 59 | 99 |
| | | Reverse: TGAAGCGTGGACATGATCCG | 60 | |
| RRM2 | ribonucleotide reductase regulatory subunit M2 | Forward: GGGAATCCCTGAAACCCGAG | 60 | 70 |
| | | Reverse: CCATCGCTTGCTGCAAAGAA | 59.7 | |
| CCNA2 | cyclin A2 | Forward: GGATGGTAGTTTTGAGTCACCAC | 59 | 202 |
| | | Reverse: CACGAGGATAGCTCTCATACTGT | 59 | |
| MAD2L1 | mitotic arrest deficient 2 like 1 | Forward: ACGGTGACATTTCTGCCACT | 60 | 105 |
| | | Reverse: TGGTCCCGACTCTTCCCATT | 60.5 | |
| HPRT1 | hypoxanthine phosphoribosyl-transferase 1 | Forward: GGACTAATTATGGACAGGACTG | 55 | 195 |
| | | Reverse: GCTCTTCAGTCTGATAAAATCTAC | 55 | |

were considered as statistically significant when $P < 0.05$ ($^*P < 0.05$, $^{**}P < 0.01$, $^{***}P < 0.001$).

## Results

### 1 Identification of DEGs in SCLC

117 SCLC samples and 51 normal lung samples were involved in this study (Table 1 and S1 Table). There were 3337 DEGs (1752 upregulated and 1585 down-regulated) in GSE40275, 510 DEGs (326 up-regulated and 184 down-regulated) in GSE99316-GPL570, and 2304 DEGs (953 up-regulated and 1351 down-regulated) in GSE60052 which were identified between SCLC tissues and normal lung tissues as shown in volcano plots (Fig 2A–2C). The Venn diagram analysis of these DEGs mapped that 208 DEGs, including 102 up-regulated genes and 106 down-regulated genes, were consistently found in the three data sets (Fig 2D–2E and S2 Table). All 208 DEGs are listed in Table 3. As shown in Fig 2F, we screened top 20 differentially expressed up-regulated and down-regulated genes respectively by the cut-off criteria.

Chromosome mapping of DEGs presented gene distribution on chromosomes, with chromosomes 1 containing the most dysregulated genes in SCLC (Fig 2G). Interestingly, four genes showed dysregulation on the X chromosome in SCLC (*FHL1*, *SRPX*, *HMGB3* and *WNK3*), while no genes on the Y chromosome was affected.

### 2 Functional annotation and pathway enrichment analysis

Funrich, as a tool for the analysis of genes and proteins, was used for GO functional annotation and biological pathway enrichment analysis of DEGs.

There are three categories, including biological process (BP), cellular component (CC) and molecular function (MF), involved in GO functional annotation. In Fig 3, the top 10 enriched GO projects were displayed. Analysis of GO BP suggested that up-regulated DEGs were significantly enriched in cell cycle, regulation of nucleobase, nucleoside, nucleotide and nucleic acid

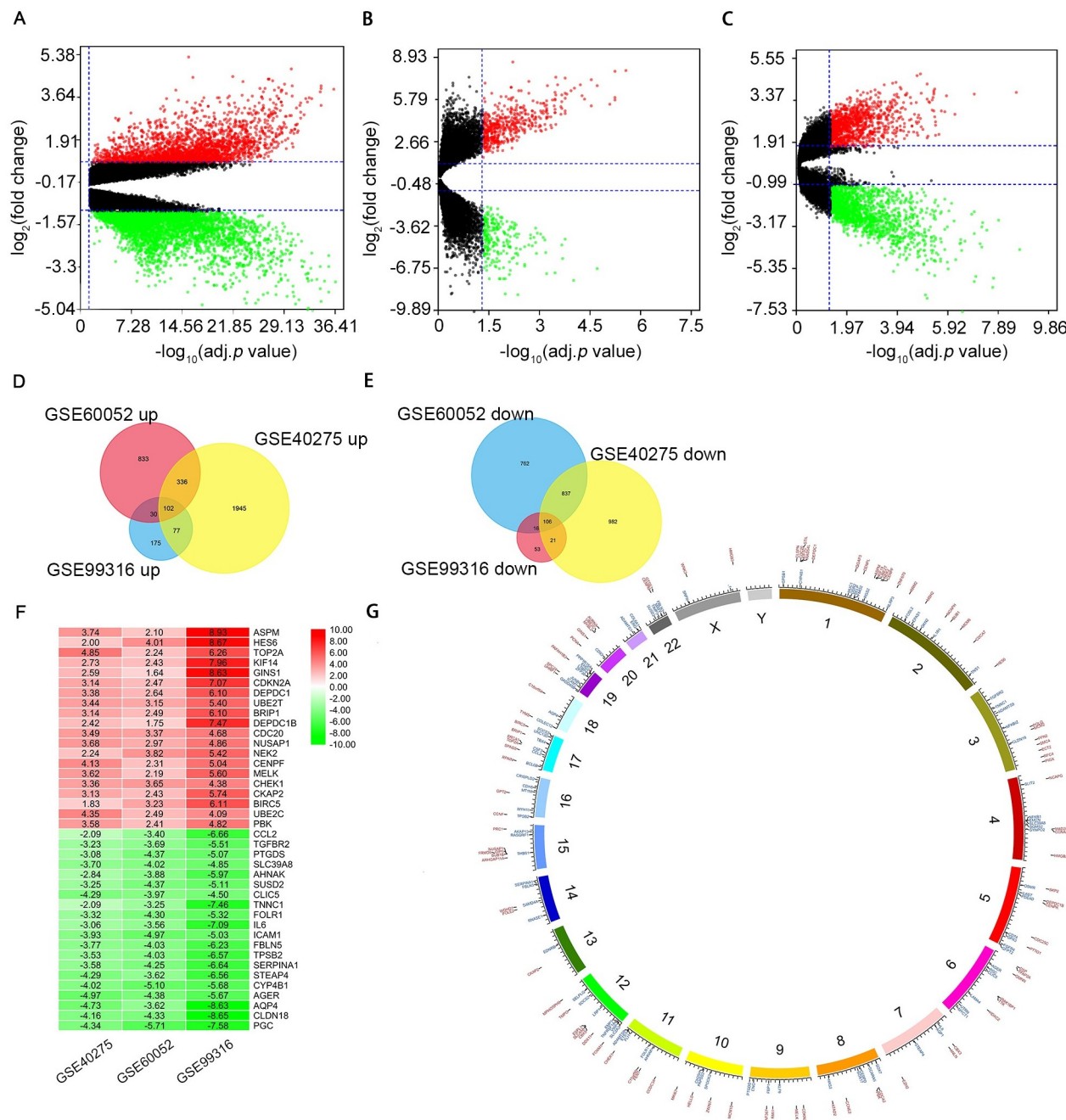

**Fig 2. Identification of differentially expressed genes (DEGs) between SCLC and normal tissues.** Volcano plots displayed the number of DEGs verified from the three GEO datasets. (A) volcano plots of GSE40275, (B) volcano plots of GSE99316, (C) volcano plots of GSE60052. Green denotes a lower expression level, red denotes a higher expression level and black denotes that the gene has no different according to the cut-off criteria. Venn diagram demonstrates the overlapping DEGs, including (D) 102 upregulated and (E) 108 downregulated genes among the three datasets. (F) Heat map of top20 DEGs for upregulated and downregulated genes. Green denotes a lower expression level, red denotes a higher expression level and black denotes that the gene has no different according to the cut-off criteria. Each column denotes a dataset and each row denotes a gene. The number in each rectangle denotes a normalized gene expression level. (G) Chromosome mapping of DEGs. Blue denotes down-regulated DGEs, and red denotes up-regulated DGEs.

**Table 3. Two hundred and eight differentially expressed genes (DEGs) were identified and confirmed from three GEO datasets, including 102 up-regulated genes and 108 down-regulated genes in SCLC tissues, compared to normal lung tissues.**

| Regulation | Number | DEGs (Gene symbol) |
|---|---|---|
| **Up-regulated** | 102 | *TOP2A,UBE2C,CENPF,PRC1,ASPM,NUSAP1,EZH2,CCNE2,HELLS,MELK,PBK, CDC20,TYMS,ARHGAP11A,UBE2T,DEPDC1,KIF2C,CHEK1,KIAA0101,MKI67,BRIP1, CDKN2A,CKAP2,HMGB3,ANLN,NCAPG,PTTG1,RFC4,MSH2,BUB1B,CENPK, ATAD2,TTK,PFN2,BUB1,CCNA2,RRM2,AURKA,ECT2,KIF14,WDHD1,HMGB2,STIL, PCNA,KIAA1524,MPHOSPH9,GINS1,UHRF1,ZWINT,SPAG5,DEPDC1B,GMNN, RMI1,MPO,POLE2,DSP,NEK2,MCM6,MCM2,CLSPN,SMC4,WNK3,CBX3,FOXM1, POLQ,HES6,CENPL,BRCA1,CDCA7,CDC25C,CDCA2,HDAC2,BIRC5,MAD2L1, C11orf80,RAD54L,IQGAP3,SKP2,FEN1,CBX5,PIGX,MYBL2,NCAPH,CCDC34, PAFAH1B3,GTSE1,ESPL1,GPT2,SLC4A8,TFAP2A,ZNF670,CCNF,OIP5,MCM10, C18orf54,CENPM,RPAIN,FRMD5,IRAK1BP1,ZNF367,DDX11,SPC24* |
| **Down-regulated** | 106 | *NFKB1,SOX17,DDR2,NLRP3,SPSB1,WISP2,GFPT2,MAFF,PDE4D,COL6A1,IL6ST, BCL6B,AKAP13,SOX7,ETS1,SLC2A3,CRISPLD2,CSF3,SOCS2,CH25H,UNC13D, SAMD4A,TNFRSF1A,SCARA5,ADAMTS8,KLF9,NFKBIZ,RASGRF1,FLI1,AHNAK, FBLN1,SRPX,SYNPO2,EHD2,THBS1,FOSL2,LAMA4,DAMTS9,GUCY1A3,LTBP4, FGFR4,CEBPD,TNNC1,CCL2,LRP1,TBX4,SELPLG,ERG,SLIT2,RNASE1,JUNB,HAS2, TMEM2,MYCT1,PPP1R15A,GADD45B,TIMP3,SELP,OSMR,RGS2,CD74,UTRN,TNS1, PTRF,COLEC12,EMCN,ENG,MT1M,MYH11,EMP1,SPOCK2,CD93,MUC1,SOCS3, IL1R1,FHL1,CDH5,FBP1,VWF,IL6,PTGDS,ZFP36,ADAMTS1,TGFBR2,SUSD2,PAPSS2, SGMS2,FOLR1,EDNRB,TPSB2,GKN2,SERPINA1,AQP1,SLC39A8,FBLN5,FMO2,GPX3, ICAM1,CYP4B1,EPAS1,CLDN18,CLIC5,STEAP4,PGC,AQP4,AGER* |

metabolism (Fig 3A). For analysis of CC, the genes were significantly enriched in nucleus, nucleoplasm, kinetochore, and chromosome (Fig 3C). The analysis of MF for these genes mainly included DNA binding (Fig 3E).

Analysis of GO BP indicated that down-regulated DEGs were most but not significantly enriched in cell growth and/or maintenance, and transmembrane receptor protein tyrosine kinase signaling pathway (Fig 3B). For analysis of CC, the genes were manifestly enriched in extracellular, extracellular space, and extracellular matrix (Fig 3D). Finally, analysis of MF of these genes showed that they were apparently enriched in extracellular matrix structural constituents (Fig 3F).

Furthermore, pathway enrichment analysis was carried out for up-regulation and down-regulation DEGs. Candidate genes of up-regulated DEGs were mainly enriched in mitotic cell cycle/ M-M/G1 phases, DNA replication, and ATM pathway (Fig 4A and Table 4). Moreover, a critical gene *CDC20* was particularly enriched in cell cycle and DNA replication in pathway enrichment analysis for up-regulated genes (Table 4 and S3 Table). The notably enriched pathways for down-regulated DEGs were epithelial-to-mesenchymal transition, amb2 integrin signaling, and interleukin-6 (*IL-6*) signaling pathway (Fig 4B). However, Interleukin-6 (*IL-6*), a crucial gene associated with inflammation was significantly enriched in amb2 Integrin signaling, Interleukin-mediated signaling, and cytokine signaling in the immune system in pathway enrichment analysis for down-regulated genes (Table 4 and S3 Table).

## 3 Analysis of PPI network and modules

As results, the PPI network showed that a total of 178 nodes and 2466 protein pairs were acquired with a score > 0.4. The main nodes in this network were the up-regulated DEGs (Fig 5A and S4 Table). Furthermore, two modules (module 1 and module 2) with score >4 were detected by MCODE (Fig 5B–5C). All nodes in module 1 with an MCODE score of 56.094 (65 nodes, 1795edges) were up-regulated DEGs, while all nodes in modules 2 with an MCODE score of 4.60 (11nodes, 23edges) were down-regulated DEGs in SCLC samples.

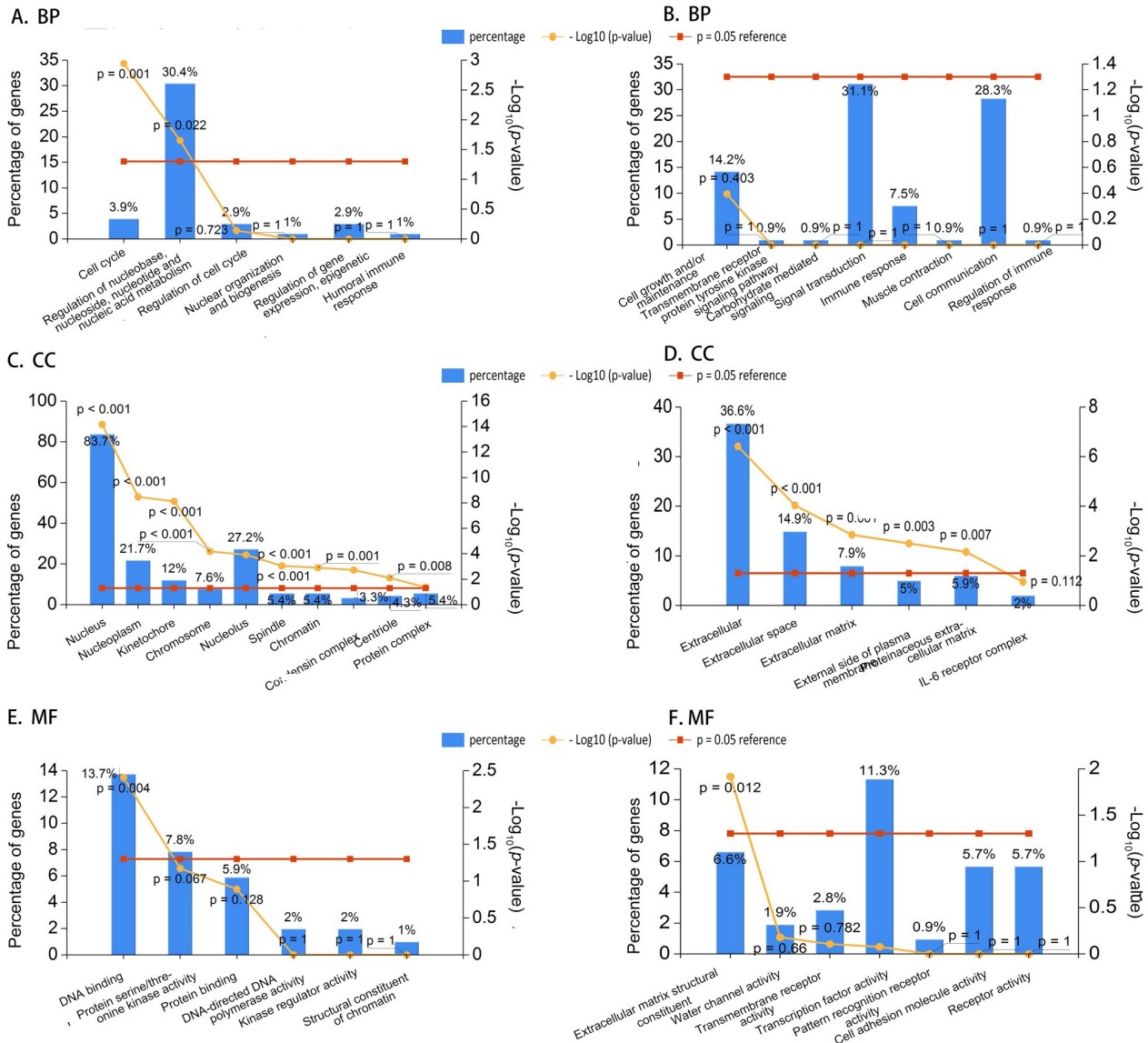

**Fig 3. Functional annotation and pathway enrichment analysis for DGEs.** (A) BP enrichment of up-regulated DEGs; (B) BP enrichment of down-regulated DEGs; (C) CC enrichment of up-regulated DEGs; (D) CC enrichment of down-regulated DEGs; (E) MF enrichment of up-regulated DEGs; (F) MF enrichment of down-regulated DEGs. X axis denotes detailed appellations of functional annotation and pathway enrichment; Y axis denotes percentage of genes or–log10(*P*-value). Red line denotes *P* = 0.05, the reference value for cut-off criteria of statistical analysis; Yellow line denotes–log10(*P*-value).

Furthermore, the biological pathways enrichment analysis of two modules were shown in Fig 6 and S5 Table. The mitotic cell cycle pathway was identified as the most significant pathway in module 1 (Fig 6A), and amb2 Integrin signaling was the most significant pathway in module 2 (Fig 6B).

## 4 Hub genes screening

For seeking hub genes in the PPI networks, all these node pairs were identified by cytohubba. As shown in Table 5, the genes, *CDC20*, *BUB1*, *TOP2A*, *RRM2*, *CCNA2*, *UBE2C*, *MAD2L1*,

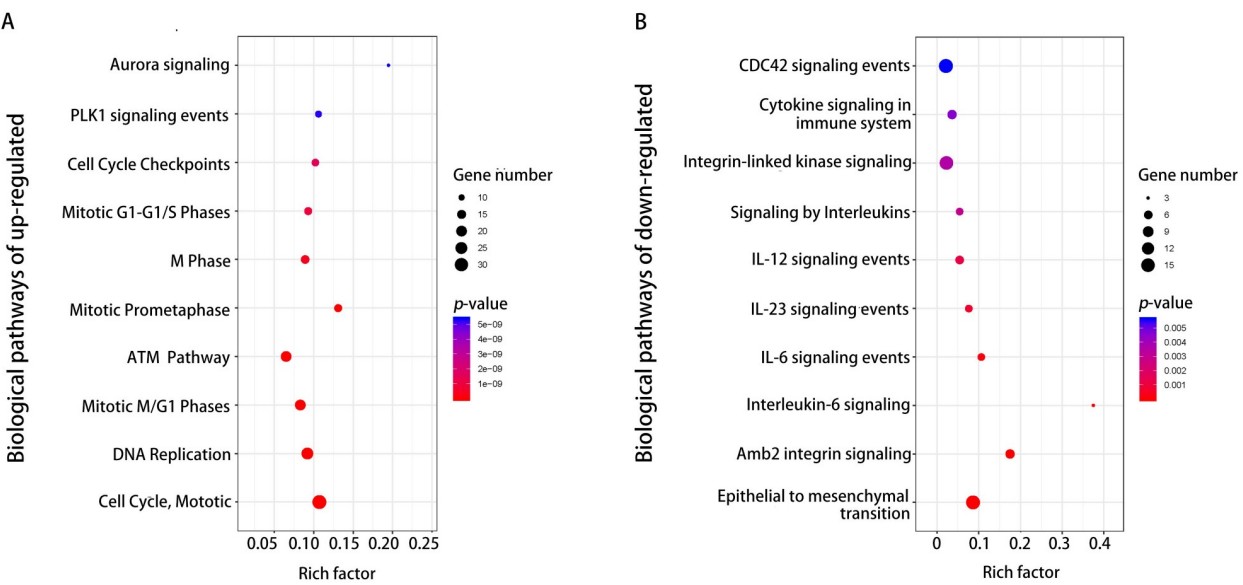

**Fig 4. Enriched biological pathways of up-regulated DEGs (A) and down-regulated DEGs (B).**

and *BUB1B*, were hub genes with higher node degrees, and they were all up-regulated genes in module 1.

## 5 Verification of hub gene mRNA expression levels

Firstly, the mRNA expression levels of *CDC20*, *BUB1*, *TOP2A*, *RRM2*, *CCNA2*, *UBE2C*, *MAD2L1*, and *BUB1B* were significantly increased in SCLC samples compared with normal lung samples based on the Oncomine database, which was consistent with the above bioinformatics investigation (Fig 7). Secondly, further study was conducted to verify the expression levels of these hub genes by RT-qPCR through tissue samples of SCLC patients and paired adjacent non-cancerous ones. The mRNA levels of eight hub genes in SCLC tissues were significantly overexpressed compared to those in paired adjacent ones. (Fig 8).

## 6 Analysis of hub gene-drug interaction network

CTD was used to study the interaction between hub genes and available therapeutic drugs of cancer. As results, multiple drugs could alter the expression of these eight hub genes, including

**Table 4. The top 10 enriched biological pathways of up-regulated and down-regulated DEGs.**

| Term of up-regulated DEGs | Overlap | *P*-value | Term of down-regulated DEGs | Overlap | *P*-value |
|---|---|---|---|---|---|
| Cell Cycle, Mitotic | 34/317 | 5.58E-28 | Epithelial-to-mesenchymal transition | 16/185 | 6.33E-11 |
| DNA Replication | 24/261 | 1.36E-17 | amb2 Integrin signaling | 7/40 | 1.77E-07 |
| Mitotic M-M/G1 phases | 20/242 | 1E-13 | Interleukin-6 signaling | 3/8 | 6.51E-05 |
| ATM pathway | 20/307 | 8.88E-12 | IL6-mediated signaling events | 5/47 | 0.000137 |
| Mitotic Prometaphase | 13/99 | 1.1E-11 | IL23-mediated signaling events | 5/66 | 0.000683 |
| M Phase | 14/158 | 3.62E-10 | IL12-mediated signaling events | 6/111 | 0.001187 |
| Mitotic G1-G1/S phases | 13/140 | 9.35E-10 | Signaling by Interleukins | 5/92 | 0.003028 |
| Cell Cycle Checkpoints | 12/118 | 1.56E-09 | Integrin-linked kinase signaling | 15/154 | 0.00356 |
| PLK1 signaling events | 11/104 | 5.3E-09 | Cytokine Signaling in Immune system | 7/193 | 0.004573 |
| Aurora B signaling | 8/41 | 5.38E-09 | CDC42 signaling events | 16/755 | 0.005622 |

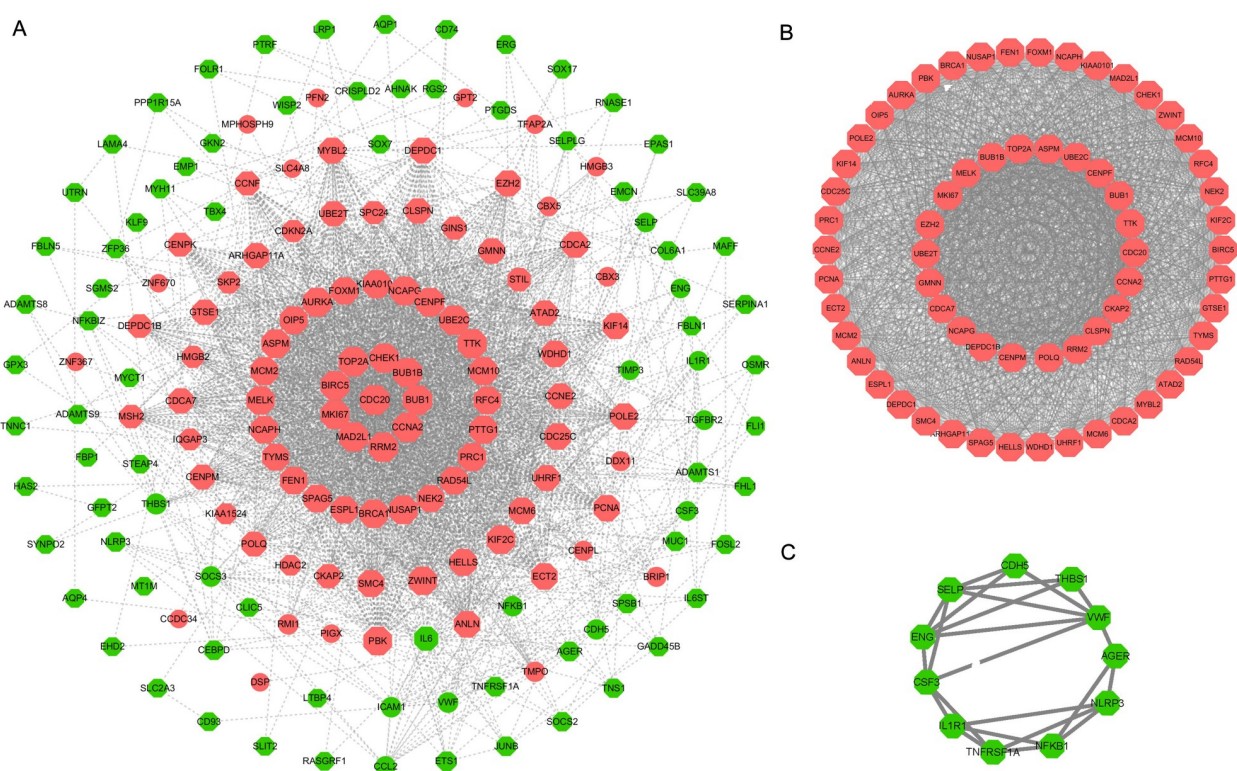

**Fig 5. Construction and analysis of PPI network.** (A) PPI network constructed by the DEGs of three GEO datasets. The significant modules, module1(B) and module2(C), identified from the PPI network by MCODE method. The red nodes denote upregulated DEGs, and the green nodes denote downregulated DEGs.

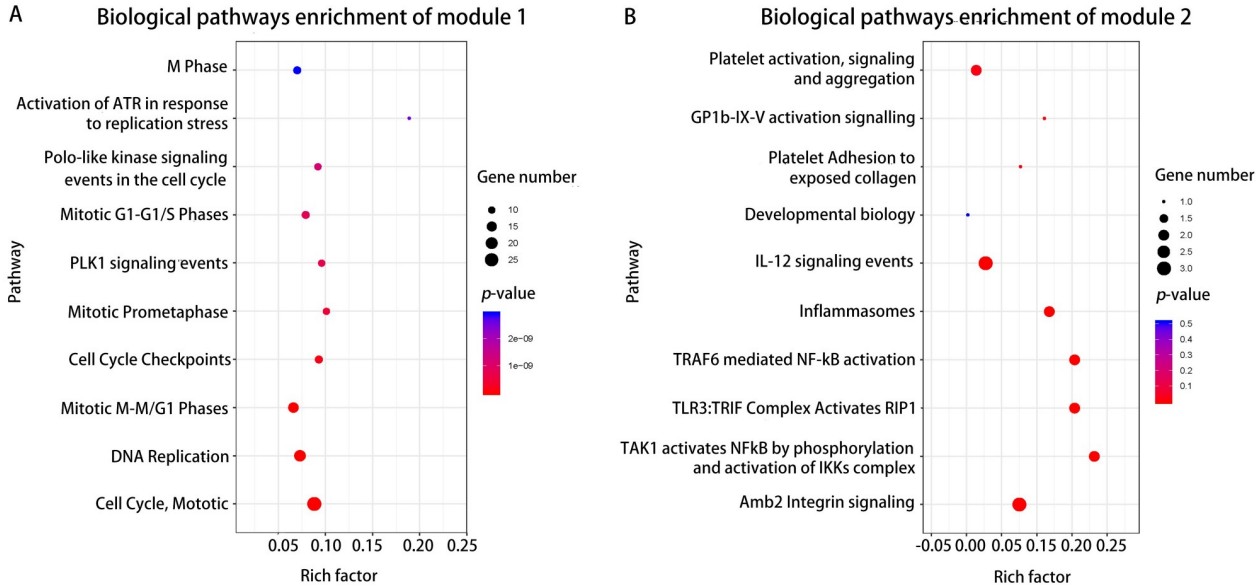

**Fig 6. Biological pathway enrichment analysis of two modules.** The top 10 biological pathways enrichment analysis of DEGs in module 1(A) and module 2(B) by FunRich, respectively, P < 0.05. X axis denotes rich factor, namely the enrichment levels; Y axis denotes the top 10 significantly enriched pathways of two modules. The color of the dot indicates the different *P*-value, and the size of the dot denotes the number of candidate genes enriched in the corresponding pathway.

**Table 5. Hub genes with high node degrees.**

| Gene symbol | Degree score | Type | MCODE cluster |
|---|---|---|---|
| CDC20 | 78 | Up-regulated | Cluster 1 |
| BUB1 | 77 | Up-regulated | Cluster 1 |
| TOP2A | 77 | Up-regulated | Cluster 1 |
| RRM2 | 76 | Up-regulated | Cluster 1 |
| CCNA2 | 76 | Up-regulated | Cluster 1 |
| UBE2C | 76 | Up-regulated | Cluster 1 |
| MAD2L1 | 75 | Up-regulated | Cluster 1 |
| BUB1B | 75 | Up-regulated | Cluster 1 |

*CDC20*, *BUB1*, *TOP2A*, *RRM2*, *CCNA2*, *UBE2C*, *MAD2L1*, and *BUB1B*. For example, Suniti-nib, Methotrexate and Fluorouracil could inhibit the expression of *CDC20* while Irinotecan could promote the expression of *CDC20* (Fig 9A).

## Discussion

Due to the insufficiency in effective targeted therapy options, SCLC is considered a "neglected sibling" compared to NSCLC. The mutations of epidermal growth factor receptor (*EGFR*)

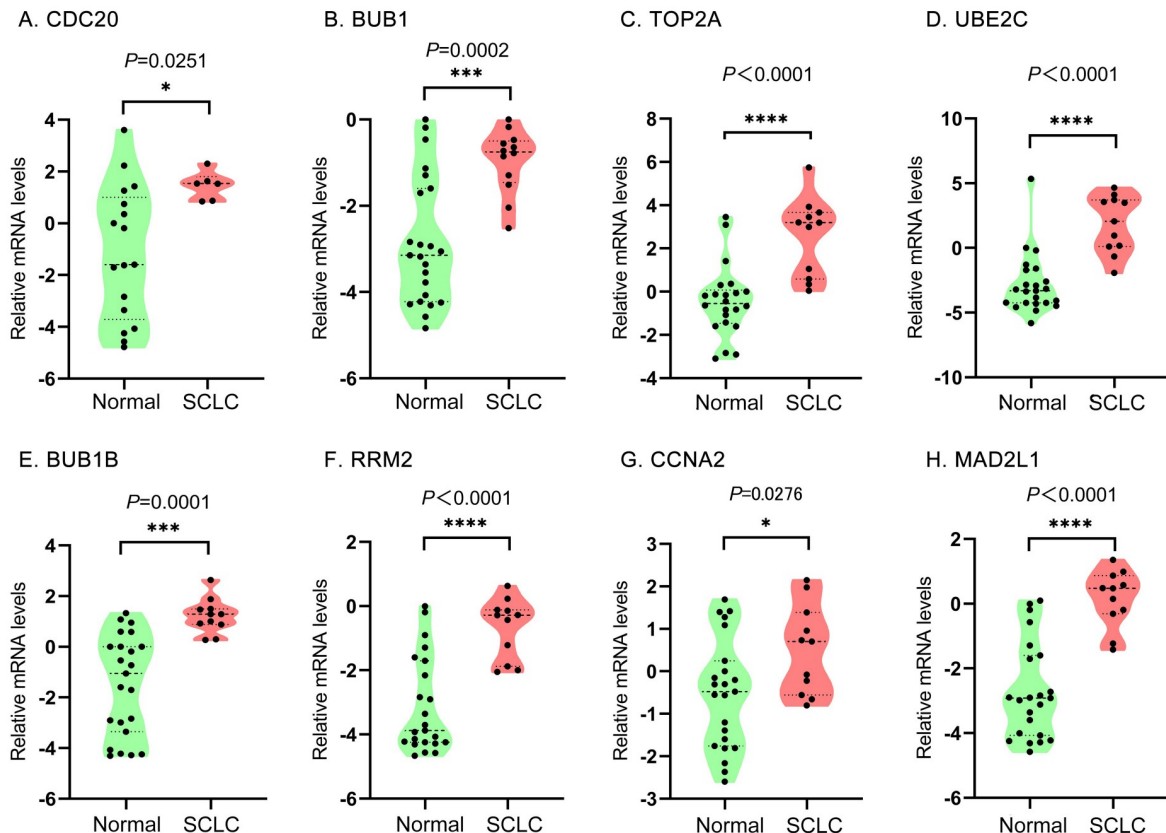

**Fig 7. Analysis of expression of hub genes in the Oncomine database.** Relative mRNA expression levels of (A) CDC20, (B) BUB1, (C) TOP2A, (D) UBE2C, (E) BUB1B, (F) RRM2, (G) CCNA2, and (H) MAD2L1 in normal lung and SCLC samples. The expression level data were standardized by $\log_2$ conversion and median centered. Data are presented as violin plot with minimum (from bottom to top), 25th percentile, median, 75th percentile, and maximum. The black dots denote the sample size. Data were analyzed using paired student's t-test. Differences were considered as statistically significant when $P < 0.05$ (* $P < 0.05$, ** $P < 0.01$, *** $P < 0.001$).

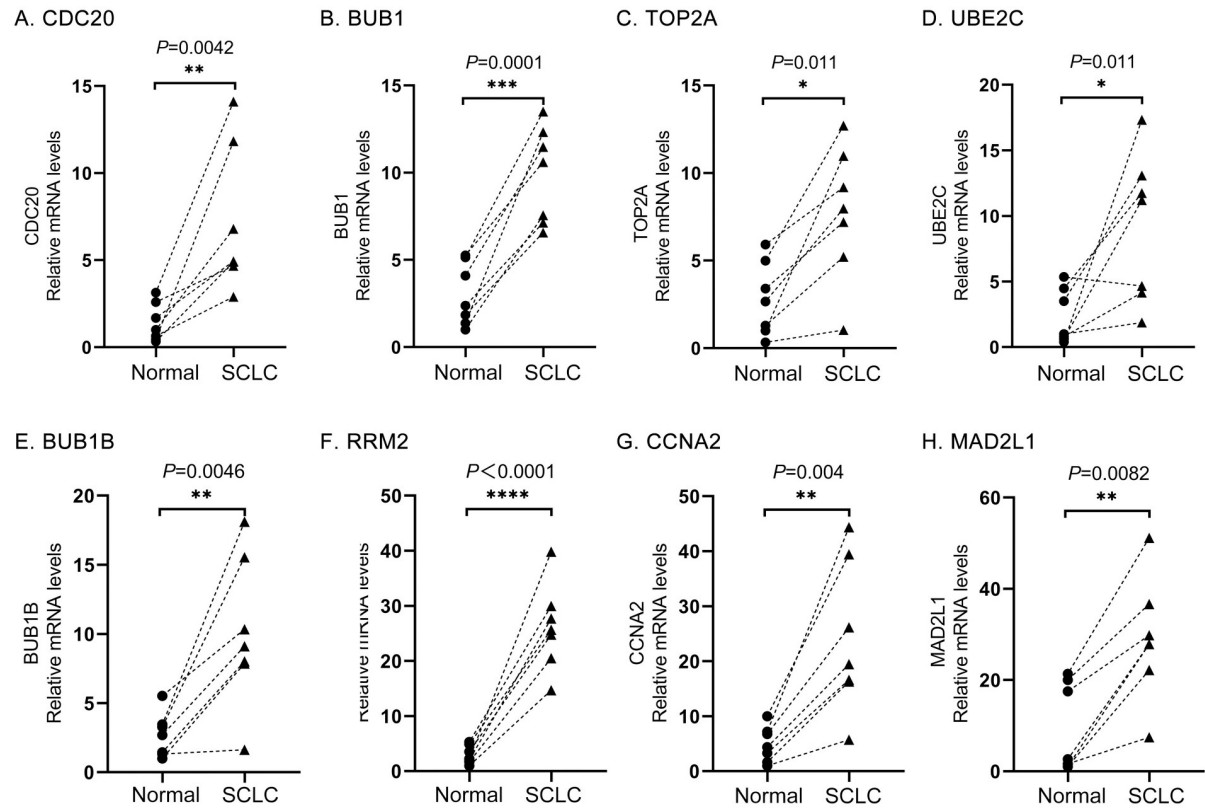

**Fig 8. The relative expression of eight hub genes in human SCLC samples (n = 7) and paired adjacent non-cancerous tissue samples (n = 7) were detected using RT-qPCR.** (A) CDC20, (B) BUB1, (C) TOP2A, (D) UBE2C, (E) BUB1B, (F) RRM2, (G) CCNA2, and (H) MAD2L1. HPRT1 was internal reference gene.analyze.

kinase, the fusion of the anaplastic lymphoma kinase (*ALK*), and ROS Proto-Oncogene 1 Receptor Tyrosine Kinase (*ROS1*), result in an immense improvement in the remedy of sufferers with NSCLC [25–28]. However, these genetic variations are not common in SCLC, and targeted drugs for NSCLC are not effective for SCLC [29]. Therefore, novel biomarkers with high efficiency, high sensitivity and high specificity are urgently needed for diagnosis and prognosis of SCLC.

Compared with single array analysis, multiple arrays integration is considered as a better method to enhance detection capabilities and improve the reliability of results [30]. In this study, we analyzed three SCLC data sets from GEO to gain insight into gene expression patterns on a genome-wide scale. Then, 208 DEGs (102 up-regulated and 108 down-regulated) and eight hub genes have been identified and used for further analysis. Chromosome mapping of 208 DEGs displayed that chromosome 1 contained the most dysregulated genes in SCLC. Previously studies affirmed that early-stage development of lung cancer was associated with X chromosome inactivation in females. The inactivation test of X-chromosome could be used to screen women who are prone to malignant tumors, including lung cancer [31]. Our findings suggested that the abnormal expression of *FHL1*, *SRPX*, *HMGB3* and *WNK3* on the X chromosome may be related to SCLC in females [32]. However, there is no differential expression gene in Y chromosome in our present study.

A PPI network was constructed for the purpose of predicting the protein functional association of 208 identified DEGs. As a result, up-regulated genes were predominantly enriched in mitotic cell cycle, DNA replication, mitotic M-M/G1 phases, and ATM pathway in SCLC.

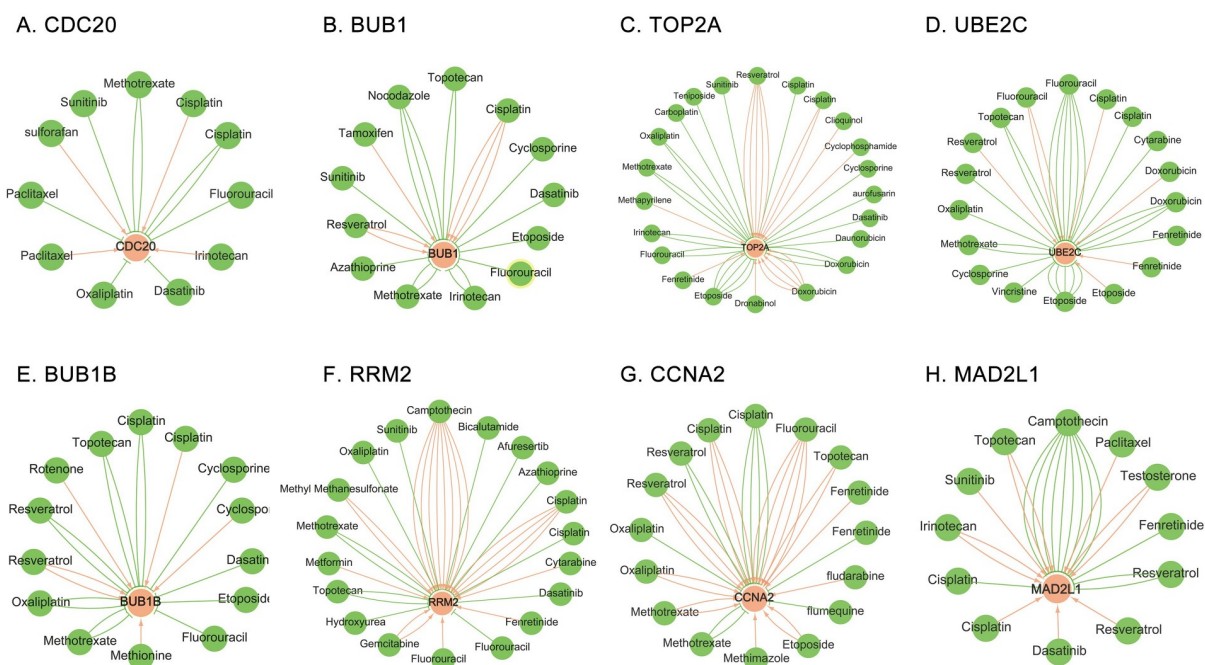

**Fig 9. Construction of gene-drug interaction network among eight hub genes and corresponding available chemotherapeutic drugs.** The available chemotherapeutic drugs could induce or reduce the expression level of these hub genes, including (A) CDC20, (B) BUB1, (C) TOP2A, (D) UBE2C, (E) BUB1B, (F) RRM2, (G) CCNA2, and (H) MAD2L1, in mRNA or protein. The red arrow denotes that chemotherapy drugs promote the expression of hub gene; the green arrow indicates that chemotherapy drugs inhibit the expression of hub gene. In this network, the number of arrows between drugs and genes represents the number of supports in previous literature.

Meanwhile, down-regulated genes were mainly enriched in epithelial-to-mesenchymal transition, amb2 Integrin signaling, and *IL-6* signaling pathway in SCLC. Among the top 10 biological pathways enrichment analysis for down-regulated genes, 9 pathways were closely related to IL-6 and immune system. (Table 4 and S3 Table) It indicated that the immune level of SCLC was relatively low, and application of IL-6 inhibitors, immunotherapy or activation of immunity might be the potential strategies for the treatment of SCLC.

In the current study, the eight hub genes selected by cytohubba were overexpressed in SCLC tissues compared to paired adjacent non-cancerous tissues. Lastly, seven of those hub genes, namely *CDC20, TOP2A, BUBI, BUBIB, UBE2C, CCNA2* and *MAD2L1* (all *P*-values <0.05), were closely related to the cell cycle pathway. Additionally, the mRNA expression of hub genes were searched for by mining the Oncomine database and human samples, which further validated the bioinformatics analysis. Although previous studies have authenticated that most of these disordered hub genes were closely related to the diagnosis, treatment and prognosis of various diseases, the precise functions and molecular mechanism of them in the occurrence and development of SCLC have not yet been clearly illuminated.

As an evolutionarily conservative process, cell cycle plays an imperative role in cell growth and differentiation. Dysregulations of cell cycle are considered as a hallmark of human cancer [33]. In the treatment of cancer, many strategies for the cell cycle have been implemented. Increasingly studies have revealed that several genes related to cell cycle such as *CDC20, TOP2A, BUBI, BUBIB*, and *UBE2C*, are related to the occurrence and development of cancer, which were also identified in the present study.

Cell division cycle protein 20 (*CDC20*), locating in chromosome 1, is a promoter for the anaphase-promoting complex (APC). Early research has demonstrated that *CDC20* is highly

expressed in multiple tumors, and is related to the poor prognosis of patients with gastric cancer, liver cancer, bladder cancer, colon cancer [34], and NSCLC [35]. Subsequent studies have indicated that the inhibition of CDC20 expression could reduce the colony formation rate of lung cancer cells in NSCLC and SCLC [36]. Therefore, *CDC20* may be viewed as a novel or latent target for the amelioration of SCLC.

Topoisomerase II alpha (*TOP2A*) gene, encoding a 170 kDa nuclear enzyme that regulates the DNA topological state during the process of DNA transcription and replication. It catalyzes the fracture and reconnection of double stranded DNA, thus changing the topological structure of DNA [37]. Numerous studies indicated that *TOP2A* is highly expressed in a variety of malignant tumors, such as colorectal cancer [38], meningioma [39], breast cancer [40], adrenocortical carcinoma [41], NSCLC [42] and SCLC [43, 44]. Topoisomerase II Inhibitors have been illustrated as active agents in SCLC cell lines [45, 46], and levels of *TOP2A* are important determinants of drug response in SCLC [47]. In our current study, *TOP2A* was overexpressed in SCLC and mostly enriched in mitotic cell cycle pathway. Consequently, the results of our study are in concert with those of previous studies, suggesting that *TOP2A* may be a direct or indirect factor in the occurrence and deterioration of SCLC

The multidomain protein kinases *BUB1* (aliases: *BUB1A*) and *BUB1B* (aliases: *BUBR1*) are key elements of the mitotic checkpoint for spindle assembly [48]. *BUB1* plays an important role in chromosome assembly and kinetochore localization in cells [49–51]. While *BUB1B* is associated with stabilizing centromere-microtubule junction and chromosome alignment [52]. Upregulation of *BUB1B* can prevent aneuploidy and cancer and prolong healthy lifetime [53]. Abnormal expression of *BUB1* and *BUB1B* were resulted in the prognosis of patients with brain tumor [54], glioblastoma [55], colorectal cancer [56], and NSCLC [57], and resulted in the impairment of mitotic checkpoint function. Therefore, future studies on *BUB1* and *BUB1B* combining genetic approaches may provide an effective strategy for clinical anti-tumor treatment and could be the focus of SCLC research in the future.

As reported, ubiquitin-conjugating enzyme E2C (UBE2C), belonging to the E2 ubiquitin-conjugating enzyme family, plays crucial roles in a variety of malignancies, namely breast cancer [58], colorectal cancer [59], melanoma [60], and hepatocellular carcinoma [61]. Otherwise a study declared that poorer OS of NSCLC patients were related to *UBE2C* overexpression [62]. Upregulation of *UBE2C*-mediated autophagy promoted NSCLC progression [63].

Ribonucleotide reductase regulatory subunit M2 (*RRM2*) is a significant enzyme in DNA replication. High expression of *RRM2* was uncovered in glioblastoma [64] with promoting tumorigenicity [65], prostate cancer [66], NSCLC [67], and breast cancer. Overexpression of *RRM2* was strongly associated with worse survival in breast cancer and increased expression was shown in tamoxifen resistant patients [68]. Recently, a study [69] showed that chemotherapy resistance was associated with RRM2/EGFR/AKT signaling pathway in NSCLC.

Cyclin A2 (*CCNA2*), belonging to cyclin family, is a regulator of cell cycle. It activates cyclin dependent kinase 2(CK2) and promotes transformation through G1/S and G2/M [70]. *CCNA2* was overexpressed in bladder cancer [71], ER+ breast cancer and related to tamoxifen resistance [72], hepatoma with promoting cell proliferation [73] and lung adenocarcinoma [74]. *CCNA2* might have prognostic value for progression free survival(PFS) and OS in patients with lung cancer [75].

As an integral part of the mitotic spindle assembly checkpoint, mitotic arrest defect 2 like 1 (*MAD2L1*) is manifestly enriched in the cell cycle pathway and ensures that all chromosomes are arranged correctly on the metaphase plate [76, 77]. The deletion of tumor suppressor *MAD2L1* could lead to premature degradation of cyclin B, mitosis failure in human cells and tumorigenesis [78]. *MAD2L1* is overexpressed in multiple cancerous, such as breast cancer [79, 80] and gastric cancer [81], and lung cancer [82]. Recent studies [83–85] demonstrated

that the prognosis of SCLC patients with high *MAD2L1* expression is worse than that with low *MAD2L1* expression. It implies that *MAD2L1* may be a promising therapeutic target for SCLC.

The interaction between eight hub genes and anti-tumor drugs were analyzed for the better understanding of the possibility of these genes as promising therapeutic targets for SCLC. As a result, we discovered that multiple drugs could change the expression of these hub genes.

As shown in Fig 9A, Sulforafan and Irinotecan promoted the expression of *CDC20*, while Sunitinib, Methotrexate, Fluorouracil, Dasatinib and Oxaliplatin inhibited the expression of *CDC20*. This suggested that Sunitinib, Methotrexate, Fluorouracil, Dasatinib, and Oxaliplatin might be viewed as targeted drug for the treatment of SCLC patients with high expression of *CDC20*. Interestingly, the interaction between Cisplatin or Paclitaxel and *CDC20* were controversial, and the conclusions drawn in different studies were inconsistent. In the same way, Sunitinib, Nocodazole, Topotecan, Cyclosporine, Dasatinib, Etoposide, Fluorouracil, Irinotecan, Methotrexate, and Azathioprine might be viewed as targeted drugs for the treatment of SCLC patients with high expression of BUB1. (Fig 9B) Methotrexate, Oxaliplatin, Carboplatin, Teniposide, Sunitinib, Cyclosporine, Aurofusarin, Dasatinib, Daunorubicin, Dronabinol, Etoposide, Fluorouracil, and Irinotecan might be viewed as targeted drugs for the treatment of SCLC patients with high expression of TOP2A. (Fig 9C) Vincristine, Cyclosporine, Methotrexate, Oxaliplatin, Topotecan and Cytarabine might be viewed as targeted drugs for the treatment of SCLC patients with high expression of UBE2C. (Fig 9D) Methotrexate, Oxaliplatin, Topotecan, Dasatinib, Etoposide and Fluorouracil might be viewed as targeted drugs for the treatment of SCLC patients with high expression of BUB1B. (Fig 9E) Methotrexate, Oxaliplatin, Sunitinib, Bicalutamide, Afureserti, Azathioprine, Dasatinib, Hydroxyurea and Topotecan might be viewed as targeted drugs for the treatment of SCLC patients with high expression of RRM2. (Fig 9F) Camptothecin and Fenretinide might be viewed as targeted drugs for the treatment of SCLC patients with high expression of MAD2L1. (Fig 9H) Surprisingly, no confirmed one could be viewed as targeted drug for the treatment of SCLC patients with high expression of CCNA2. (Fig 9G) However, further experimental study, including *in vivo* and *in vitro* experiments and clinical studies, are needed to explore the relationship between these hub genes and the prognosis of SCLC patients, and whether SCLC patients can benefit from these drugs.

At present, some related studies on the SCLC core genes have been published. Liao et al screened five hub genes from four GEO datasets (GSE60052, GSE43346, GSE15240 and GSE6044) by the raw data analysis by R software, functional annotation and pathway enrichment analysis, PPI network analysis, enrichment analyzes of two significant modules, and analysis of the expression levels of hub genes in the Oncomine database [84]. Wen et al discerned 10 hub genes from two GEO databases (GSE6044 and GSE11969) by using GEO2R tool, GO functional annotation and KEGG pathway enrichment analysis, PPI network, module analysis, hub genes selection, and validation of the mRNA expression levels of hub genes in the Oncomine database [86]. Mao et al identified 19 hub genes and 32 miRNAs from two GEO datasets (GSE6044 and GSE19945). Further analysis performed by functional annotation and pathway enrichment analysis of DEGs, PPI network, module analysis, hub genes screening, and miRNA-gene regulatory network [87]. Compared with the published literatures, we all found DEGs at RNA level based on GEO database and analyzed by bioinformatics methods. We constructed functional annotation and pathway enrichment analysis, PPI network, enrichment analyzes of significant modules, and verification of the expression levels of hub genes in the Oncomine database of DEGs by using similar methods. But we excluded two GEO datasets, including GSE6044 and GSE11969 which were in above published reports. The reasons are as follows: As reported, the human genome contains about 20,000 to 4,5000 genes encoding proteins [88]. But the number of probes in GSE6044 was only 8,793 genes, a lot less than 20,000~50,000 genes in that three GEO datasets. Therefore, the inadequacy of these data may

lead to incomplete analysis. Secondly, the database related technology platform was applied 10 more years ago and had not been updated recently, therefore GSE6044 was ruled out. For GSE11969, all expression data with $\log_2$ (fold change) is less than 2, which is completely different from other GEO data sets. It may be unreasonable. Therefore, GSE11969 was excluded. More advantages of this study were as follows: Firstly, RT-qPCR were performed to validate the mRNA expression levels of hub genes by human samples. Secondly, this study screened potential prognostic biomarkers or therapeutic targets in SCLC and performed hub gene-drug interaction network analysis.

In the present paper, we have discussed that the overexpression of eight hub genes was closely related to the occurrence and development of SCLC, indicating that these hub genes might be acted as promising prognostic markers or therapeutic targets for SCLC. But our research also has limitations. Firstly, the data utilized in this study were all collected from public databases, but the quality of the data cannot be evaluated. Secondly, the sample capacity of relevant data is comparatively small. Thirdly, our study focused only on genes that changed significantly in multiple data sets, the characteristics of race, region, gender, age, tumor classification, stage, and smoking status were not considered integrally. Therefore, a lot of valuable biological information may be ignored in our research. Finally, as results, all eight hub genes were overexpressed in SCLC, but the corresponding mechanism has not been fully elucidated. Therefore, more molecular evidence is needed. Moreover, the current research in SCLC lacked prognostic data related to these hub genes, such as survival curves, which brought limitations to the clinical application value of hub genes. In this paper, the expression levels of eight hub genes were mainly analyzed. Whether these hub genes could be used as biomarkers or therapeutic targets of SCLC required further study.

## Conclusions

In conclusion, our bioinformatics analysis identified 208 DEGs, eight hub genes (*CDC20*, *BUB1*, *TOP2A*, *RRM2*, *CCNA2*, *UBE2C*, *MAD2L1*), and the mitotic cell cycle pathway that might play an momentous role in the development and prognosis of SCLC. As shown in database analysis and confirmed by human samples, overexpression of these hub genes indicated a poor prognosis for patients with SCLC. These results indicated that a comprehensive study of these DGEs will help us to understand the pathogenesis and progression of SCLC. However, since this study is mainly based on data analysis, further basic mechanism studies and clinical studies are needed to confirm these hypotheses in SCLC. We hope that this study can furnish certain new genomic basis for the individualized treatment of SCLC.

## Supporting information

**S1 Table. Gene expression microarray datasets for SCLC in GEO datasets, including GSE40275, GSE99316, and GSE60052.**
(XLSX)

**S2 Table. Information for the DEGs identified from the GEO datasets (|log2FC| $\geq$ 1, adjust *P* value $<$ 0.05).**
(XLSX)

**S3 Table. Enrichment analysis of the DEGs.**
(XLSX)

**S4 Table. PPI network analysis of the DEGs.**
(XLSX)

**S5 Table. Pathway enrichment analysis of the DEGs in the two modules.**
(XLSX)

## Acknowledgments

Very grateful to my colleague, Yinyin Wang, for the help of collecting clinical samples during the experiment.

## Author Contributions

**Data curation:** Yugang Huang.

**Formal analysis:** Li Wang, Sen-yuan Luo, Yugang Huang.

**Funding acquisition:** Xiaomin Su.

**Investigation:** Sen-yuan Luo.

**Methodology:** Li Wang, Xiaomin Su, Yugang Huang.

**Project administration:** Xianbin Tang, Yugang Huang.

**Validation:** Li Wang.

**Visualization:** Xiuwen Chen.

**Writing – original draft:** Xiuwen Chen, Xiaomin Su, Xianbin Tang, Yugang Huang.

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
