## [Decision Letter · Decision Letter 0]

9 Oct 2020

PONE-D-20-29317

Identification of Potential Crucial Genes and Key Pathways in Small Cell Lung Cancer Based on Integrated Bioinformatic Analysis and Human Samples

PLOS ONE

Dear Dr. Huang,

Thank you for submitting your manuscript to PLOS ONE. After careful consideration, we feel that it has merit but does not fully meet PLOS ONE’s publication criteria as it currently stands. Therefore, we invite you to submit a revised version of the manuscript that addresses the points raised during the review process.

We look forward to receiving your revised manuscript.

Kind regards,

Sumitra Deb, PhD

Academic Editor

PLOS ONE

Journal Requirements:

2. Thank you for submitting the above manuscript to PLOS ONE. During our internal evaluation of the manuscript, we found significant text overlap between your submission and the following previously published work.

https://www.frontiersin.org/articles/10.3389/fgene.2018.00469/full

Please revise the manuscript to rephrase the duplicated text, cite your sources, and provide details as to how the current manuscript advances on previous work. Please note that further consideration is dependent on the submission of a manuscript that addresses these concerns about the overlap in text with published work.

Reviewers' comments:

Reviewer's Responses to Questions

**Comments to the Author**

1. Is the manuscript technically sound, and do the data support the conclusions?

Reviewer #1: Yes

Reviewer #2: Yes

2. Has the statistical analysis been performed appropriately and rigorously? 

Reviewer #1: Yes

Reviewer #2: Yes

3. Have the authors made all data underlying the findings in their manuscript fully available?

Reviewer #1: Yes

Reviewer #2: Yes

4. Is the manuscript presented in an intelligible fashion and written in standard English?

Reviewer #1: Yes

Reviewer #2: Yes

5. Review Comments to the Author

Reviewer #1: Huang et al. explored the potential pathogenic and prognostic crucial genes and key pathways of SCLC via bioinformatics analysis of public datasets. The authors identified eight hub genes that could be new biomarkers for prognosis and treatment for SCLC. Since there are several publications that similarly analyze and identify hub genes in SCLC, the authors need to discuss in depth about commonality and difference between the current and past analyses. The authors also need to discuss further the interpretation and significance of the Gene-Drug Interaction analysis in Fig. 9.

Reviewer #2: In this study the authors have data mined three high-throughput gene expression data sets to assess differentially expressed genes in small cell lung cancer (SCLC) and corelated how over -expression of certain genes can be used to determine therapeutic strategy. Differentially expressed genes between normal and SCLC tissues which overlapped among in the 3 data sets were identifies and top 20 genes (upregulated and down regulated) were further screened. Pathway enrichment analysis showed that a most of the upregulated genes were associated with the cell cycle, DNA replication pathway whereas the down regulated genes were primarily involved in interleukin-mediated signaling, and cytokine signaling. They also performed gene expression studies in SCLC samples and identified 8 hub genes. Through gene-drug interaction they showed how certain chemotherapeutic drugs could affect the expression of these genes.

The manuscript is well written, and the authors have acknowledged the limitations of their study in discussion section.

Minor Revisions:

1) Figure 2. Legend. (F) should be capitalized.

2) Figure 3. The figure legend should be more detailed as to what the red and yellow lines are denoting. It will make the data descriptive and informative.

3) The authors saw that the downregulated genes were mainly associated with immune signaling. The authors have not discussed this observation in the discussion section. It would be informative if the authors discussed about what is known and what are the implications.

6. PLOS authors have the option to publish the peer review history of their article (what does this mean?). If published, this will include your full peer review and any attached files.

Reviewer #1: No

Reviewer #2: No

---

## [Author Response · Author response to Decision Letter 0]

15 Oct 2020

Dear editors and reviewers：

On behalf of my co-author, we thank you very much for giving us an opportunity to revise our manuscript, we greatly appreciate editor and reviewers for their positive and constructive comments and suggestions on our manuscript entitled "Identification of potential target genes and crucial pathways in small cell lung cancer based on bioinformatic strategy and human samples" (the title has been renewed; Paper ID : PONE-D-20-29317). We have finished the proof reading and checking carefully, and some corrections about the proof and answers to the questions are provided below.

Corrections:

(1) The manuscript was revised by us from the title to the " References " and made according to the style requirements of PLOS ONE. 

Moreover, Dr. Su was asked to rewrite the manuscript word by word. The revised manuscript is attached. Detailed information of Dr. Su: 

Name: Xiao-min Su, 

E-mail: xiaominsu@nankai.edu.cn;

Affiliated institution: Department of Immunology, Nankai University School of Medicine, Nankai University, Tianjin 300071, China. 

Address: No.94, Weijin Road, Nankai, Tianjin 300071, China. 

Mobile phone: +86-18630934876

List of published articles in recent years:

1. Cao Shuisong#, Su Xiaomin#, Zeng Benhua#, et al. The Gut Epithelial Receptor LRRC19Promotes the Recruitment of Immune Cells and Gut Inflammation. Cell Rep, 2016, 14(4):695-707.

2. Su Xiaomin#, Yan Hui#, Huang Yugang#, et al. Expression of FABP4, adipsin and adiponectin in Paneth cells ismodulated by gut Lactobacillus. Sci Rep, 2015, 5:18588-18588.

3. Su Xiaomin#, Mei Shiyue, Liang Xue, et al. Epigenetically modulated LRRC33 actsas a negativephysiological regulator for multiple Toll-like receptors. J Leukoc Biol, 2014, 96(1):17-26.

4. Su Xiaomin#, Min Siping#, Cao Shuisong#, et al. LRRC19 expressed in the kidney

induces TRAF2/6-mediated signals to prevent infection by uropathogenicbacteria. Nat

Commun, 2014, 5:4434-4434.

(2) The title of manuscript has been renewed as "Identification of potential target genes and crucial pathways in small cell lung cancer based on bioinformatic strategy and human samples".

(3) The brief title of manuscript has been renewed as " Potential target genes and crucial pathway in SCLC".

(4) According to the Journal Requirements, we have corrected the naming of supporting information and manuscript.

(5) Ethical statements written in any section besides the " Materials and Methods "have been deleted.

Answers to questions from reviewers:

(1) Comments of reviewer #1: Huang et al. explored the potential pathogenic and prognostic crucial genes and key pathways of SCLC via bioinformatics analysis of public datasets. The authors identified eight hub genes that could be new biomarkers for prognosis and treatment for SCLC. Since there are several publications that similarly analyze and identify hub genes in SCLC, the authors need to discuss in depth about commonality and difference between the current and past analyses. The authors also need to discuss further the interpretation and significance of the Gene-Drug Interaction analysis in Fig. 9.

Answer to Reviewer #1: 

Firstly, Due to the similar analysis and identification of hub genes in SCLC in some publications, we discuss the similarities and differences between current and past analyses. Please see the 432nd line to the 444th line of the revised manuscript.

Secondly, we further discussed the interpretation and significance of the gene drug interaction analysis in Fig. 9. Please see the 396th to 417th line of the revised manuscript.

(2) Minor Revisions 1: Figure 2. Legend. (F) should be capitalized.

Answer to Minor Revisions 1: In Figure 2 legend, (F) has been capitalized. Please see the 213rd line of the revised manuscript.

(3) Minor Revisions 2: Figure 3. The figure legend should be more detailed as to what the red and yellow lines are denoting. It will make the data descriptive and informative.

Answer to Minor Revisions 2: In Figure3, the figure legend has been detailed as to what the X axis, Y axis, red and yellow lines are denoting. Please see the 213rd line to the 216th line of the revised manuscript.

(4) Minor Revisions 3: The authors saw that the downregulated genes were mainly associated with immune signaling. The authors have not discussed this observation in the discussion section. It would be informative if the authors discussed about what is known and what are the implications.

Answer to Minor Revisions 3: This study found that the down regulated genes were mainly related to immune signals. We further discussed this observation in the discussion section. Please see the 318th line to the 322nd line of the revised manuscript

Lastly, we have upload figure files to the Preflight Analysis and Conversion Engine (PACE) digital diagnostic tool to ensure these figures meet PLOS requirements.

We deeply appreciate the efficient, professional and rapid processing of our paper by your team. If there is anything else wo should do, please don't hesitate to contact us at the address below.

Thank you and best regards.

Your sincerely,

Yu-gang Huang

Corresponding author: 

Name: Yu-gang Huang; E-mail: huangyg2018@outlook.com

---

## [Decision Letter · Decision Letter 1]

29 Oct 2020

Identification of potential target genes and crucial pathways in small cell lung cancer based on bioinformatic strategy and human samples

PONE-D-20-29317R1

Dear Dr. Huang,

We’re pleased to inform you that your manuscript has been judged scientifically suitable for publication and will be formally accepted for publication once it meets all outstanding technical requirements.

Kind regards,

Sumitra Deb, PhD

Academic Editor

PLOS ONE

Additional Editor Comments (optional):

Reviewers' comments:

Reviewer's Responses to Questions

**Comments to the Author**

1. If the authors have adequately addressed your comments raised in a previous round of review and you feel that this manuscript is now acceptable for publication, you may indicate that here to bypass the “Comments to the Author” section, enter your conflict of interest statement in the “Confidential to Editor” section, and submit your "Accept" recommendation.

Reviewer #1: All comments have been addressed

Reviewer #2: All comments have been addressed

2. Is the manuscript technically sound, and do the data support the conclusions?

Reviewer #1: (No Response)

Reviewer #2: Yes

3. Has the statistical analysis been performed appropriately and rigorously? 

Reviewer #1: (No Response)

Reviewer #2: Yes

4. Have the authors made all data underlying the findings in their manuscript fully available?

Reviewer #1: (No Response)

Reviewer #2: Yes

5. Is the manuscript presented in an intelligible fashion and written in standard English?

Reviewer #1: (No Response)

Reviewer #2: Yes

6. Review Comments to the Author

Reviewer #1: (No Response)

Reviewer #2: (No Response)

7. PLOS authors have the option to publish the peer review history of their article (what does this mean?). If published, this will include your full peer review and any attached files.

Reviewer #1: No

Reviewer #2: No

---

## [Editor Report · Acceptance letter]

5 Nov 2020

PONE-D-20-29317R1 

Identification of potential target genes and crucial pathways in small cell lung cancer based on bioinformatic strategy and human samples 

Dear Dr. Huang:

I'm pleased to inform you that your manuscript has been deemed suitable for publication in PLOS ONE. Congratulations! Your manuscript is now with our production department. 

Kind regards, 

on behalf of

Dr. Sumitra Deb 

Academic Editor

PLOS ONE